# Altered Cord Blood Lipid Concentrations Correlate with Birth Weight and Doppler Velocimetry of Fetal Vessels in Human Fetal Growth Restriction Pregnancies

**DOI:** 10.3390/cells11193110

**Published:** 2022-10-02

**Authors:** Stephanie S. Chassen, Karin Zemski-Berry, Stefanie Raymond-Whish, Camille Driver, John C. Hobbins, Theresa L. Powell

**Affiliations:** 1Department of Pediatrics, Section of Neonatology, University of Colorado Anschutz Medical Campus, Aurora, CO 80045, USA; 2Division of Endocrinology, Metabolism & Diabetes, University of Colorado Anschutz Medical Campus, Aurora, CO 80045, USA; 3Department of Obstetrics & Gynecology, Division of Maternal Fetal Medicine, University of Colorado Anschutz Medical Campus, Aurora, CO 80045, USA; 4Department of Obstetrics & Gynecology, Division of Reproductive Sciences, University of Colorado Anschutz Medical Campus, Aurora, CO 80045, USA

**Keywords:** fetal growth restriction, long-chain polyunsaturated fatty acids, placenta lipids, Doppler velocimetry, cord plasma

## Abstract

Fetal growth restriction (FGR) is associated with short- and long-term morbidity, often with fetal compromise in utero, evidenced by abnormal Doppler velocimetry of fetal vessels. Lipids are vital for growth and development, but metabolism in FGR pregnancy, where fetuses do not grow to full genetic potential, is poorly understood. We hypothesize that triglyceride concentrations are increased in placentas and that important complex lipids are reduced in cord plasma from pregnancies producing the smallest babies (birth weight < 5%) and correlate with ultrasound Dopplers. Dopplers (umbilical artery, UA; middle cerebral artery, MCA) were assessed longitudinally in pregnancies diagnosed with estimated fetal weight (EFW) < 10% at ≥29 weeks gestation. For a subset of enrolled women, placentas and cord blood were collected at delivery, fatty acids were extracted and targeted lipid class analysis (triglyceride, TG; phosphatidylcholine, PC; lysophosphatidylcholine, LPC; eicosanoid) performed by LCMS. For this sub-analysis, participants were categorized as FGR (Fenton birth weight, BW ≤ 5%) or SGA “controls” (Fenton BW > 5%). FGRs (n = 8) delivered 1 week earlier (*p* = 0.04), were 29% smaller (*p* = 0.002), and had 133% higher UA pulsatility index (PI, *p* = 0.02) than SGAs (n = 12). FGR plasma TG, free arachidonic acid (AA), and several eicosanoids were increased (*p* < 0.05); docosahexaenoic acid (DHA)-LPC was decreased (*p* < 0.01). Plasma TG correlated inversely with BW (*p* < 0.05). Plasma EET, non-esterified AA, and DHA correlated inversely with BW and directly with UA PI (*p* < 0.05). Placental DHA-PC and AA-PC correlated directly with MCA PI (*p* < 0.05). In fetuses initially referred for inadequate fetal growth (EFW < 10%), those with BW ≤ 5% demonstrated distinctly different cord plasma lipid profiles than those with BW > 5%, which correlated with Doppler PIs. This provides new insights into fetal lipidomic response to the FGR in utero environment. The impact of these changes on specific processes of growth and development (particularly fetal brain) have not been elucidated, but the relationship with Doppler PI may provide additional context for FGR surveillance, and a more targeted approach to nutritional management of these infants.

## 1. Introduction

The human fetus relies on the uteroplacental unit for adequate delivery of oxygen and nutrients for optimal growth during its 40-week gestation period. Circumstances whereby a fetus fails to achieve its genetic growth potential, most commonly due to uteroplacental insufficiency, results in fetal growth restriction (FGR). The consequences of a gestation complicated by FGR are plentiful and can have a marked impact on an infant’s short- and long-term health. In the short-term, FGR is associated with significant neonatal morbidities including those related to prematurity, poor thermoregulation, feeding intolerance, organ dysfunction, and sepsis, to name a few [1]. In the long term, FGR fetuses have an increased risk of metabolic (obesity, insulin resistance, diabetes), cardiovascular, and neurodevelopmental (cognitive, behavioral, motor skills) impairments that can span the life course [2,3].

The most common cause of FGR is uteroplacental insufficiency, characterized by impaired uteroplacental and umbilical blood flow. Normal placental development involves remodeling of the maternal spiral arteries, which supply maternal blood to the intervillous space, bathing the fetal-derived capillary networks (chorionic villi) lined by the transporting epithelium (syncytiotrophoblast). This remodeling is stimulated by endovascular trophoblast invasion of the arterial walls, partial replacement of the maternal endothelium, and loss of arterial smooth muscle. The subsequent spiral artery dilation allows for a high flow, low resistance vessel to perfuse the intervillous space with maternal blood. In cases of limited or absent endovascular trophoblast invasion of these arteries, there is minimal vessel dilation leading to a high resistance conduit with persistent smooth muscle. This insufficient remodeling predisposes to hypoperfusion, hypoxia, oxidative stress and villous maldevelopment, and is characteristic of pathologic pregnancy conditions like FGR and preeclampsia [4,5].

Doppler velocimetry is a mainstay of fetal surveillance in high-risk pregnancies to follow changes in fetal circulation and optimize timing of delivery to avoid serious fetal compromise. Reduced end-diastolic umbilical artery (UA) velocity is associated with increased vascular resistance in the fetoplacental circulation as evidenced by increased UA pulsatility index (PI) and correlates with adverse pregnancy outcomes including perinatal death [6,7,8]. Worsening in utero conditions result in redistribution of blood flow away from fetal peripheral vascular beds towards vital organs such as the brain, as a fetal adaptive response [9,10,11,12,13,14]. This preferential blood supply to the brain (“brain sparing”) is evidenced on Doppler by *decreased* resistance to flow (decreased PI values) in the middle cerebral artery (MCA). The ratio of MCA PI/UA PI constitutes the cerebroplacental ratio (CPR), which reflects the status of small vessel circulation in the placenta. CPR is considered a more comprehensive Doppler index for predicting perinatal outcome, as it is representative of both the placental status and fetal adaptation through circulatory redistribution [15].

FGR can be classified based on gestational age at diagnosis. Expert consensus opinion defines early onset FGR as diagnosis at <32 weeks and late onset as diagnosis at ≥32 weeks [16]. In addition to age of onset these groups differ in clinical manifestations, patterns of fetal deterioration, severity of placental dysfunction, and association with hypertensive disorders of pregnancy [17]. Early onset FGR is classically associated with a lack of spiral artery remodeling and thus is more severe, demonstrates evidence of fetal cardiovascular adaptation on Doppler, and is more often associated with maternal hypertensive disorders of pregnancy. Late onset is more common, but generally associated with milder placental dysfunction and less fetal hemodynamic adaptation.

A significant consequence of impaired uteroplacental and umbilical blood flow in FGR is altered placental function, including transport of critical nutrients from mother to fetus, such as amino acids [18,19,20], sodium/potassium [21,22], glucose [19,23], and calcium [24], as reported by our group and others. Placental transport of fatty acids, critical for development of the brain and retina, is generally less well understood; however, our group has previously demonstrated an upregulation of placental fatty acid transport proteins and an increase in long chain polyunsaturated fatty acids (LCPUFA) stored as triglycerides in human FGR placentas [25].

Fatty acids serve several vital biological roles in fetal development. They provide energy for cell metabolism and fetal growth and act as building blocks for the formation of complex lipids with varying functions. These complex lipids include triglycerides (storage), phospholipids (cell membrane integrity and central nervous system fatty acid transfer, particularly phosphatidylcholine), and eicosanoids (cell signaling). In pregnancy, fetal fat deposition increases exponentially across gestation, reaching its maximal accretion at term. This coincides with physiologic changes promoting placental fatty acid transfer, such as increased placental exchange surface area and blood flow, maturation of the placental villi (site of maternal-fetal nutrient exchange) and increased maternal circulating non-esterified fatty acids and triglycerides [26]. FGR fetuses are at risk for markedly decreased subcutaneous fat depots, often depending on severity of growth restriction, which can have important consequences for early postnatal life since these fat stores are a large source of essential lipids in the first months of extrauterine life [27,28].

LCPUFA are vital for brain and retina development, especially docosahexaenoic acid (DHA) and arachidonic acid. Importantly, these LCPUFAs must be transferred across the placenta from maternal circulation, as the activity of desaturation and elongation enzymes necessary for LCPUFA synthesis is low in both the placenta and fetus [29]. Adequate placental transport and robust accumulation in fetal fat depots are therefore paramount to normal brain development.

This study aimed to determine concentrations of fatty acids and complex lipid molecules in placenta and umbilical cord blood from FGR and small for gestational age (SGA) controls, and to determine relationships between these levels and measures of fetal growth and wellbeing. We hypothesized that FGR placentas have an increased storage of LCPUFA in triglycerides, subsequently reducing circulating LCPUFA-containing complex lipid molecules in FGR cord blood, and correlate with markers of poor fetal wellbeing measured by ultrasound.

## 2. Materials and Methods

### 2.1. Study Subjects and Antenatal Assessments

Women were recruited following referral to outpatient centers associated with the University of Colorado between 2015–2019. The study was approved by the Colorado Multiple Institutional Review Board (IRB number 14-1360) and informed consent was obtained from all study participants. Women were included in this prospective observational case–control study if pregnant with a singleton fetus whose estimated fetal weight (EFW) was less than the 10th percentile for gestational age based on Hadlock growth curves [30] in the 2nd or 3rd trimester, were greater than 18 years of age, had early ultrasound dating (<20 weeks), and had no evidence of fetal or chromosomal anomalies. Ultrasound imaging (Voluson E10, GE Healthcare, Zipf, Austria) was performed every 1 to 4 weeks from study recruitment depending on gestational age and FGR severity. Though study recruitment took place at multiple University of Colorado referral sites, only those women that delivered at the University of Colorado Hospital and for whom both placentas and cord blood were collected were included in this study analysis. This study utilized “intention to treat” methodology. Therefore, enrolled participants remained in the study even if on occasion, their EFW on longitudinal scans strayed above the 10th percentile. Ultrasound measurements included EFW, abdominal circumference, and Doppler PI of fetal vessels (UA, MCA). CPR was calculated and defined as MCA PI divided by UA PI [31]. EFW was calculated using measurements of the biparietal diameter, head and abdominal circumference, and femur length according to the Hadlock equation [30]. Measurements from the last ultrasound prior to delivery were used for this study analysis.

### 2.2. Tissue Collection and Processing

Following the delivery of women who gave birth at the University of Colorado Hospital, a placental wedge representing approximately 10% of the total weight of the intact organ was collected for immediate processing, leaving the remaining 90% available for clinical pathologic analysis. This approach was the standard operating procedure for collection of any FGR placenta for the purpose of research at our institution during the study period. This guideline was imposed by the Director of Gynecologic and Obstetric Pathology to ensure that a significant amount of placental tissue remained for clinical pathologic analysis and clinical management. Trophoblast chorionic villous tissue pieces (1 cm^3^) were dissected from the placental wedge following removal of decidua basalis and chorionic plate, washed in cold saline, and homogenized (Polytron 2500E, Kinematica AG, Bohemia, NY, USA) in Buffer D (250 mM sucrose, 1 mM Tris-HEPES, 1 mM EDTA, pH 7.4) containing protease and phosphatase inhibitors, on ice. Homogenized samples were snap frozen in liquid nitrogen and stored at −80 °C until further processing. Heparinized blood samples were obtained from the umbilical cord and processed to obtain plasma at delivery and stored at −80 °C.

### 2.3. Postnatal Assessments and Case–Control Classifications

Clinical data including maternal age, ethnicity, mode of delivery, gestational age, and infant sex was recorded at the time of delivery. Anthropometric measurements were obtained on the neonate including birth weight, length, and head circumference following birth. Investigators and official bodies have attempted to identify those undergrown fetuses at greater risk for death and morbidity from those who are “constitutionally small”. Although controversy exists regarding the usefulness of various Doppler indices, fetal size alone can provide extremely useful information about the degree of deprivation. For this reason, and due to the nature of our intent to treat methodology, we have pragmatically categorized our study population into those with birth weight ≤5th percentile (FGR “cases”) from those above this threshold (SGA “controls”) based on Fenton growth curves [32] and have chosen to analyze the Doppler indices (MCA PI, UA PI, CPR) separately.

### 2.4. Lipid Extraction of Plasma and Placenta

Fifty µL cord plasma and a total of 1 mg protein from the placental homogenate were used for lipid extraction. Starting sample volumes were brought up to a total volume of 750 µL in water, and methanol (900 µL) was added. An internal standard cocktail containing PC-19:0/19:0 (2000 pmol), d7-PC-18:1/OH (200 pmol), and 1,2,3-triheptadecanoyl glycerol (1500 pmol) was added and lipid extraction was performed by the addition of methyl-*tert*-butyl ether (3 mL) according to Matyash et al. [33]. 

#### 2.4.1. Triglyceride Analysis

For triglyceride analysis, samples were injected into an HPLC system (Shimadzu LC-10AD VP, Columbia, MD, USA) connected to a triple quadrupole mass spectrometer (Sciex 2000QTRAP, Framingham, MA, USA) and reverse phase chromatography was performed using a C18 column (50 × 3 mm, Kinetex 2.6 µm, Phenomenex). The mobile phase system consisted of solvent A (acetonitrile: H_2_O 60:40 *v*/*v*) and solvent B (isopropanol:acetonitrile 90:10 *v*/*v*) both containing 10 mM ammonium acetate. Mass spectrometric analysis was performed in the positive ion mode using multiple-reaction monitoring (MRM) of 27 triacylglycerol molecular species and the 1,2,3-triheptadecanoyl glycerol internal standard. The precursor ions monitored were the molecular ions [M + NH_4_]^+^ and the neutral loss FA product ions were monitored and summed together for each species for quantitation [34]. Concentration was determined using stable isotope dilution with standard curves for saturated and unsaturated triglycerides.

#### 2.4.2. Phospholipid Analysis

For phospholipid analysis, samples were injected into an HPLC system (Shimadzu LC-10AD VP, Columbia, MD, USA) connected to a triple quadrupole mass spectrometer (Sciex 3200, Framingham, MA, USA) and normal phase chromatography was performed using a silica column (150 × 2 mm, Luna Silica 5 µm, Phenomenex). The mobile phase system consisted of solvent A (isopropanol/hexane/water (58/40/2, *v*/*v*)) and 35% solvent B (hexane/isopropanol/water (300/400/84, *v*/*v*/*v*) both containing 10 mM ammonium acetate. Mass spectrometric analysis was performed in the negative ion mode using multiple reaction monitoring (MRM). The precursor ions monitored were the acetate adducts [M + CH_3_COO]^−^ for phosphatidylcholine (PC) and lyso-phosphatidylcholine (LPC). The product ions analyzed after collision-induced decomposition were the carboxylate anions corresponding to the acyl chains esterified to the glycerol backbone. Quantitative results were determined using stable isotope dilution with standard curves for saturated and unsaturated PC compounds.

#### 2.4.3. Eicosanoid Extraction and Analysis

Eicosanoid analysis was performed as previously described [35]. One hundred microliters of cord plasma and a total of 1 mg of protein from the placental homogenate were used for eicosanoid extraction. Briefly, after addition of a deuterated internal standard mixture, samples were added to MeOH (1:2), centrifuged, and then extracted using a solid phase extraction cartridge (Strata-X 33 µm Polymeric Reversed Phase, Phenomenex). Metabolites were eluted, dried down, reconstituted, injected onto an HPLC system (Shimadzu LC-10AD VP, Columbia, MD, USA) and separated on an HPLC column (Gemini C18, 150 × 2 mm, 5 µm, Phenomenex) directly interfaced into the electrospray source of a triple quadrupole mass spectrometer (5500 QTRAP, Sciex). Quantitation was performed using standard isotope dilution curves, as previously described [35].

### 2.5. Lipid Nomenclature

The fatty acid analysis of placenta homogenate and umbilical vein plasma consists of complex lipid molecules identified by a specific nomenclature. Triglyceride concentration results are reported as *x:y*, where *x* is the total number of carbon atoms and *y* is the total number of double bonds in the 3 fatty acyl chains esterified to the glycerol backbone. No attempt was made to determine the precise positions of the fatty acyl groups esterified at the *sn*-1 or *sn*-2 position of the glycerol backbone for the PC species reported in this paper. The diacyl PC results are reported as *x*_*y* and the ether PC species are reported as O-*x*_*y*, where *x* and *y* represent the fatty acids identified by number of carbons: number of double bonds [36]. LPC results identify the single fatty acid attached to the PC molecule following partial hydrolysis and removal of a fatty acid. Eicosanoid and non-esterified or “free” fatty acid results are also reported, with typical nomenclature used.

### 2.6. Data Presentation and Statistical Analysis

All data, including ultrasound and birth measurements, as well as placenta and plasma lipid concentrations, were analyzed for normality via visualization of histogram and QQ plot in GraphPad Prism. Statistical differences between FGR and SGA controls for normally distributed continuous variables were assessed using unpaired Student’s *t* test and presented as means [95% confidence interval]. Differences between FGR and SGA controls for non-normally distributed continuous variables were assessed using Mann- Whitney test and presented as medians [interquartile range]. Categorical variable differences between groups were assessed using Fisher’s exact test (GraphPad Prism, v9.2.0, San Diego, CA, USA). Correlations between lipid concentrations and clinical data points were assessed using Pearson’s or Spearman’s coefficient. A *p* value < 0.05 was considered statistically significant.

## 3. Results

### 3.1. Clinical Characteristics

Clinical characteristics of mother, baby, and ultrasound measurements are provided in Table 1. There were no significant differences between groups in maternal or prenatal characteristics. The majority of participants were enrolled ≥32 weeks gestation. Ultrasound growth measurements and Doppler PI of fetal blood vessels at the last ultrasound prior to delivery are reported. Notable significant differences included reduced EFW, reduced abdominal circumference percentile, and elevated UA PI in the FGR group compared to SGA controls. At birth, FGR infants were also 29% smaller in weight (with an average birth weight percentile of 2.88%), 10% shorter in length, and had a similar head circumference but a significantly increased head circumference: birth weight ratio compared to SGA controls.

### 3.2. Fatty Acid Composition

Concentrations of all measured lipid species are presented in tabular form in Appendix A (umbilical cord plasma) and Appendix A (placenta homogenate).

#### 3.2.1. Fatty Acid and Lipid Concentrations in Umbilical Vein Plasma

Umbilical vein plasma was available for analysis in all but one subject in the FGR group. FGR umbilical vein plasma demonstrated significantly increased concentrations of triglyceride molecules compared to SGA controls for nearly every triglyceride measured (Figure 1), including those with the greatest number of both carbon atoms (56) and double bonds (7), which could represent molecules containing LCPUFAs.

There were few significant differences in PC concentrations between groups (full list provided in Appendix A); however, analysis of LPCs revealed a pattern of lower concentrations in the FGR group across the measured lipid species, with significant differences in LPCs containing palmitoleic acid (16:1, Figure 2a), docosapentaenoic acid (22:5, Figure 2b), and DHA (22:6) (Figure 2c). Importantly, the levels of DHA (22:6)-LPC were 50% lower in the FGR umbilical vein.

An overall pattern of increased non-esterified fatty acid concentrations in FGR cord plasma emerged, with significantly elevated levels of palmitic (16:0), α-linolenic (18:3), oleic (18:1), stearic (18:0) and arachidonic acids (20:4) compared to SGA controls (Figure 3) but no change in DHA (22:6). A panel of arachidonic acid-derived eicosanoid molecules were analyzed (complete panel listed in Appendix A) and demonstrated significantly increased concentrations of 5- and 12-hydroxyeicosatetraenoic acid (HETE, Figure 4a), as well as all three measured epoxyeicosatrienoic acid (EET) molecules (Figure 4b) in FGR cord plasma compared to SGA controls.

#### 3.2.2. Correlations between Plasma Lipid Concentrations and Birth Weight z-Scores

Scatterplots demonstrating correlations between umbilical cord plasma lipid concentrations and birth weight z-scores are depicted in Figure 5, Figure 6 and Figure 7. Nearly all measured triglyceride species most likely to contain LCPUFAs (higher total carbon and double bond numbers) were inversely correlated with birth weight z-score (Figure 5), as were two EET eicosanoids (Figure 6). LPCs containing LCPUFA (Figure 7) were directly correlated with birth weight z-score.

#### 3.2.3. Correlations between Plasma Lipid Concentrations and Doppler Indices

Relationships between all measured lipid classes and Doppler indices were assessed. Those achieving statistical significance are presented here. LPCs containing LCPUFAs inversely correlated with UA PI (Figure 8). UA PI was also found to directly correlate with eicosanoid EET molecules (Figure 9) and non-esterified arachidonic acid and DHA (Figure 10). LCPUFA-LPCs correlated directly with CPR (Figure 11).

#### 3.2.4. Fatty Acid and Lipid Concentrations in Placenta Homogenate

There were minimal significant differences in placental homogenate concentrations of lipid molecules between groups. Appendix A demonstrate essentially no differences in triglyceride, PC, LPC or eicosanoid concentrations. Appendix A demonstrates significantly decreased concentrations of non-esterified fatty acids 16:1 (palmitoleic acid) and 18:3 (α-linolenic acid) in FGR placentas compared to control.

#### 3.2.5. Correlations between Placenta Lipid Concentrations and Clinical Characteristics

Interestingly, many PC molecules measured in placenta homogenate directly correlated with MCA PI. Nearly all measured PCs containing arachidonic acid (20:4) (Figure 12) and several PCs containing DHA (22:6) (Figure 13) demonstrated this relationship.

## 4. Discussion

Our study demonstrates that of fetuses initially referred for inadequate fetal growth (EFW < 10th percentile), those with birth weights ≤ 5th percentile (FGR) had distinctly different lipid profiles in cord blood than those with birth weights > 5th percentile (SGA controls). Many of these lipid concentrations correlated with fetal vessel Doppler PIs. To our knowledge, this is the first study to explore explicit relationships between fetoplacental lipid concentrations and specific ultrasound Doppler assessments in FGR. Our findings raise questions about fetal fatty acid metabolism including which signals may be directing incorporation of LCPUFA into specific lipid classes, and the subsequent effect on vital organ development, such as the brain.

Concentrations of several classes of lipid molecules, including nearly all measured triglycerides, several non-esterified fatty acids, and some EET and HETE isomers, were significantly elevated in cord plasma from FGR pregnancies compared to SGA pregnancies in our study. However, cord plasma concentrations of DHA (22:6)-LPC were significantly decreased in FGR. We also demonstrated correlations between lipid levels in cord plasma and placenta homogenate and markers of poor fetal wellbeing (higher UA PI, lower MCA PI, lower CPR). Specifically, our data suggest that a fetus with lower MCA Doppler velocimetry has low placental concentrations of PC containing DHA and arachidonic acid. Additionally, a fetus with higher UA PI demonstrates high levels of EET molecules, high levels of non-esterified LCPUFA (free DHA, free arachidonic acid), and low levels of LPCs containing a LCPUFA (arachidonic acid, DPA, DHA).

Differentiating pathological growth in FGR fetuses from constitutionally small fetuses is challenging. The commonly used FGR definition is ultrasonographic measurement of EFW < 10th percentile for gestational age. These babies are more likely to have severe acidosis at birth, low 5-min Apgar scores, and neonatal intensive care unit admissions [37]; however, roughly 20% will be constitutionally small without any associated morbidities [38]. Fetal weights below the 5th percentile carry a stillbirth rate as high as 2.5% [39,40], and overall the worst perinatal outcomes are observed in fetuses with EFW below the 3rd percentile or in association with abnormal Dopplers. However, nationally and internationally there is great variability in diagnostic criteria, in terms of the use of fetal biometric measurements in isolation or in combination with abnormal Doppler findings. One widely used definition for early and late FGR is based on a Delphi consensus statement from 2016 and incorporates EFW, abdominal circumference and Doppler flow of fetal vessels for diagnosis [16]. Our analyses were performed on a subset of a larger cohort of pregnancies whose inclusion criteria were an EFW < 10th percentile in 2nd or 3rd trimester. In order to identify a fatty acid phenotype in fetuses/infants most likely to have pathologic FGR consequences within this cohort, we narrowed the FGR cohort further and simplistically used birth weight of less than or equal to 5th percentile as defined by Fenton [32] to define our study group. Seven of the eight subjects in this FGR group also met the Delphi consensus group criteria for FGR diagnosis, so we do believe our findings are representative of a pathologically FGR cohort.

Previously, our group found upregulated fatty acid transport protein expression in the maternal facing plasma membrane of growth restricted human [25] and non-human primate (baboon) placentas [41], representing a presumed compensatory effort to maintain fetal fatty acid delivery in a growth restricted pregnancy. Elevated fetal lipid concentrations (non-esterified fatty acids, triglyceride containing LCPUFA, and several eicosanoids) in the current study would suggest that this placental mechanism is successful for transfer of many fatty acids.

This study demonstrated a striking increase in triglyceride concentrations in cord plasma from FGR pregnancies, a condition known to be associated with hypoxemia. Interestingly, several specific triglyceride concentrations were inversely correlated with not only birthweight, but also with MCA PI. Low values of these Doppler indices are suggestive of fetal adaptive response to hypoxemia. Previous work has demonstrated similar links between hypoxia/hypoxemia and elevated triglycerides. Mylonis et al. reported accumulation of triglycerides and lipid droplets following exposure of cultured human hepatocytes to hypoxia, mediated by hypoxia-inducible transcription factors (HIFs) [42]. Other animal studies have similarly demonstrated HIF-mediated hepatocyte lipid accumulation [43,44]. The storage of fatty acids in triglycerides and lipid droplets under hypoxic conditions may be a protective response to prevent intracellular lipotoxicity, buffering against the known lipotoxic properties of non-esterified fatty acids [42,45].

Intrapartum stress, defined as abnormal fetal heart rate pattern during labor warranting operative delivery or acidemia measured in umbilical artery at birth, can also be associated with increased fetal plasma triglycerides [46]. Patterson et al. and others have attributed this relationship to rapid mobilization of fat stores triggered by elevated catecholamines in the setting of acute stress [46,47,48]. Of the four infants in our cohort that delivered by Cesarean section, only two resulted from concern for fetal distress (non-reassuring fetal heart tones), making intrapartum/acute stress an unlikely significant contributor to our findings. We speculate that the hypoxic intrauterine environment of FGR may be the stimulus prompting the fetus to package LCPUFA (some of which possess pro-inflammatory, pro-oxidative stress properties) into triglycerides, as a protective mechanism. The correlation of fetal triglyceride concentrations with clinical ultrasound markers indicative of fetal circulatory redistribution of blood flow in the setting of adverse in utero conditions could support this. However, storage of vital LCPUFA may come at the expense of their incorporation into other, preferred forms of LCPUFA (i.e., PC and LPC) for development of vital organs such as the brain, a possible explanation for the lower DHA-LPC cord plasma levels in FGR.

Stable isotope-labeled fatty acid studies have demonstrated that the placenta preferentially takes up and transfers DHA to the fetus compared to other fatty acids [49]. The further incorporation of DHA into specific lipid classes is important for delivery to various organs. Multiple animal studies in rats and mice have determined that LPC (a PC molecule with one fatty acid removed) is the preferred carrier of DHA to the brain and retina, being more efficiently delivered than the non-esterified fatty acid, phospholipid or triglyceride form [50,51,52]. The human placenta expresses an LPC transporter (MFSD2a) on its plasma membranes, and Ferchaud-Roucher et al. found direct correlation between MFSD2a expression on the fetal-facing basal plasma membrane and DHA-LPC in the umbilical vein [53]. The availability of DHA and arachidonic acid in the brain is vital for processes including: membrane biogenesis, fluidity and function; neurogenesis and neuronal migration; synaptic process development and plasticity; modulation of receptor function, transporter and membrane-bound enzyme function; and gene expression related to signal transduction and energy metabolism [54,55,56,57,58,59]. Additionally, between the second trimester and the first two years of life there is tremendous brain growth, cell proliferation and a concomitant 60-fold increase in brain weight [60], coinciding with a 30-fold increase in DHA and arachidonic acid in the forebrain [61]. 

In FGR, susceptibility of the fetal brain to injury has been documented both antenatally and postnatally. This includes reduced total brain volume and surface area affecting both white [62,63,64,65,66] and grey matter, accompanied by cortical thinning, neuronal degeneration/death [67,68,69,70], disorganization and disrupted connectivity [71,72]. Several of these antenatal changes have been found to persist into infancy [63] and school age [73]. Overall, the placental dysfunction of FGR pregnancy is associated with a disruption of normal brain development that impacts various brain areas resulting in functional deficits (communication, problem solving, motor coordination, IQ, attention, speech, and learning) [72]. For these reasons, placental supply and delivery of DHA and arachidonic acid to the growing fetus is vital. Our study demonstrated no significant differences in placental concentrations of PC or LPC molecules containing these LCPUFA between study groups. Interestingly, nearly all placental DHA (22:6)-PC and arachidonic acid (20:4)-PC molecules directly correlated with MCA PI. This is suggestive of the effect of ongoing hypoxemia (evidenced by low MCA PI being a marker of fetal circulatory redistribution of blood flow toward the brain) on incorporation of DHA and arachidonic acid into PC. Mean placental values of these molecules were lower in FGR (Appendix A), and though not statistically significant, we speculate that a larger sample size would detect greater differences. The cord plasma DHA (22:6)-LPC levels were significantly lower in our FGR cohort and low levels correlated with lower birth weight, higher UA PI, and lower CPR—all clinically concerning indicators of an adverse in utero environment with hypoxia and redistribution of blood flow in the FGR fetus. Low cord plasma levels could be explained by placental concentrations and transport, given Ferchaud-Roucher’s work [53] as well as the correlation with clinical evidence of more severe FGR (higher UA PI) and likely effect on nutrient delivery. However, the effect of fetal triglyceride storage of LCPUFAs on DHA availability for incorporation into PC and LPC is likely a contributor. This could translate into inadequate delivery to the brain, affecting development.

Arachidonic acid is a LCPUFA that is not only vital for brain development, but also vital as a precursor to a class of bioactive lipid molecules that influence inflammation and vasoregulation called eicosanoids. Eicosanoid biosynthesis from arachidonic acid occurs via several different pathways, through the actions of three key enzyme families: COX, LOX, and CYP450. Biological functions of the different eicosanoid molecules are diverse, and frequently have opposing effects (i.e., pro- vs. anti-inflammatory, pro- vs. anti-apoptotic, vasoconstricting vs. vasodilating, etc.). The overall physiologic outcome depends on a myriad of factors including nature of the stimulus, concentrations of eicosanoids generated, and timing of production, among others [74]. FGR can be associated with increased oxidative stress [75,76] and arachidonic acid has been implicated in oxidative stress-associated pathologies, including FGR in rats [77], despite its importance in brain and retina development. Our study demonstrated correlation of low placental concentrations of arachidonic acid (20:4)-PC with low/abnormal MCA PI, suggestive of hypoxemia limiting the incorporation of arachidonic acid into PC molecules in the placenta. FGR cord plasma concentrations of non-esterified arachidonic acid and arachidonic acid (20:4)-PC molecules were increased, demonstrating a difference in fetal vs. placental arachidonic acid metabolism. Whether these fetal concentrations stem from the underlying FGR pathophysiology/oxidative stress or as a compensatory mechanism to promote brain development is currently unknown.

FGR cord plasma samples in our study also demonstrated higher concentrations of two types of arachidonic acid derivatives, HETE and EET. HETEs are classically pro-inflammatory and vasoconstricting in their actions [78,79] while EETs demonstrate anti-inflammatory, anti-apoptotic, proangiogenic, vasodilatory properties [80,81]. Higher concentrations of three EET molecules (5,6-EET; 8,9-EET; 11,12-EET) also directly correlated with higher UA PI, reflecting increased placental vascular resistance and concern for fetal wellbeing. We speculate that the hypoxic, pro-oxidative stress environment of FGR triggers the production of eicosanoids, which seems to disturb the balance of their competing functions, upon which adequate fetal growth and wellbeing in utero are dependent. Given the reciprocal functions of these two eicosanoid families, we propose that this is representative of both the underlying uteroplacental insufficiency conditions and fetal compensatory response.

The placental and cord plasma phospholipid results presented here include concentrations of both conventional diacyl PC (acyl chains attached to the sn-1 and sn-2 positions of the glycerol backbone by ester bonds) and ether PC species (alkyl chain attached to sn-1 position by an ether bond). Ether lipids constitute up to 20% of the total phospholipid pool in mammals and can contain unique structural characteristics affecting membrane fluidity/fusion/trafficking, cellular antioxidant properties, and roles in biological signaling [82]. The low placental LCPUFA-PC concentration correlation with low MCA PI in our dataset predominantly consist of ether PCs (denoted as “O-”). The degree to which the presence or absence of placental ether lipids in a pathologic in utero environment affects placental structure or antioxidant capacity remains unclear.

Our study has several limitations, including a small sample size. The Perelman IUGR Study, for which the women included in our subanalysis were recruited, enrolled women at multiple outpatient sites in the Denver Metro area affiliated with the University of Colorado. During the study period, placentas and cord blood were only collected from women who delivered at the University of Colorado Hospital. Consequently, our access to these biological samples was hindered because many subjects in the overall comprehensive study delivered at other locations. Given this small sample size, we were unable to accurately assess for the contribution of sex-based differences toward our statistically significant findings (n of just 2 males in FGR group). These would be important analyses to undertake within a larger cohort, as sex differences have been reported in the fetal response to in utero environmental stimuli [83,84], fetal adipose tissue growth and distribution [85], adult LCPUFA metabolism [86], and placental gene expression [87], to name a few. Second, our entry criteria required the EFW at the time of recruitment to be <10th percentile; however, many of the enrolled fetuses wandered above this mark. Our “intent to treat” methodology required that we continue to study these fetuses as if they were still at risk, so they were not excluded from our analyses, even those whose BW fell >10th percentile. Despite these babies not falling into the classic definition of FGR after birth, by study inclusion these infants did demonstrate that the in utero environment at some point in gestation was not ideal for completely normal growth, precluding them from being considered as “healthy” and appropriately grown for gestational age (AGA). We also know that AGA pregnancies can demonstrate abnormal fetal and placental blood vessel Doppler measurements in the setting of fetal hypoxemia [88]. In this vein, and as a consequence of study design, our third limitation is the lack of a true AGA comparison group. All pregnancies in the Perelman IUGR study were followed serially at more frequent intervals than typical gestations. Thus, a true AGA cohort could not be used as the comparison group because such detailed ultrasound measures were not available in healthy pregnancies at our institution. Despite this study limitation, the significant lipid concentration differences identified in this FGR cohort compared to SGA pregnancies lead us to believe those differences would persist (perhaps to a stronger degree) when compared to a truly AGA population. Finally, the majority of our study population was diagnosed after 32 weeks gestation. The late onset of FGR can lend itself to less severe pathology compared to those diagnosed earlier in pregnancy. The fact that we note a difference in LCPUFA metabolism in these late onset cases strengthens our paper; however, expansion of our sample size and the incorporation of more early-onset (more severe) FGR cases could strengthen our results even further.

## 5. Conclusions

In summary, our study explores fetal adaptations on a lipidomic level to FGR-associated hypoxemia and placental insufficiency. Our key findings demonstrate elevated cord plasma triglycerides, non-esterified LCPUFAs, and EETs, as well as reduced cord plasma DHA (22:6)-LPC in a small group of FGR pregnancies, all of which correlate with abnormal Doppler flow indices of fetal vessels. These relationships suggest that worsening placental vascular resistance and progression to redistribution of fetal blood flow that occurs in severe FGR influences alterations in lipid trafficking in utero. These are suspected to be fetal lipo-protective strategies (production of vasodilatory/anti-inflammatory EETs; LCPUFA storage in triglycerides) in response to the hypoxic FGR in utero environment, but these may come at the expense of appropriate trafficking and packaging of vital LCPUFA for delivery to the brain. Additional studies are needed to further delineate lipid signaling pathways directing synthesis/transport of specific lipid types in the FGR environment, as well as to evaluate alterations in LCPUFA delivery to the brain and any subsequent neurodevelopmental sequelae.

## Figures and Tables

**Figure 1 cells-11-03110-f001:**
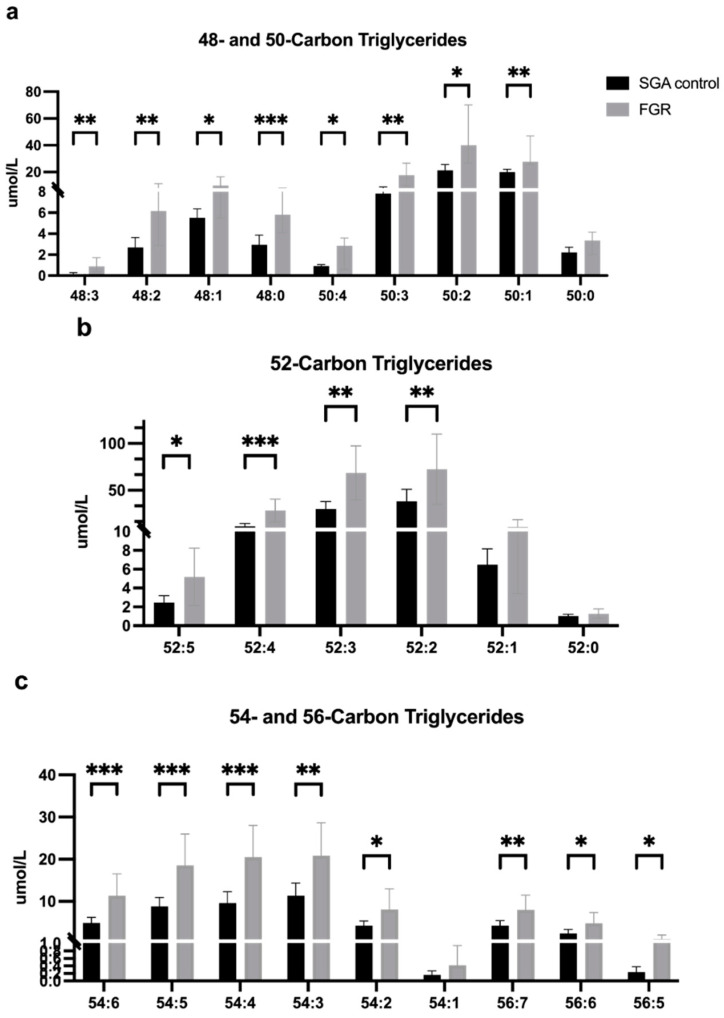
Triglyceride concentrations (μmol/L) in umbilical vein plasma. SGA control (n = 12) and FGR (n = 7) umbilical vein plasma concentrations of triglyceride molecules containing: (**a**) 48 and 50 carbon atoms; (**b**) 52 carbon atoms; and (**c**) 54 and 56 carbon atoms. Concentrations are expressed as median ± IQR for non-normally distributed data (**a**) or mean ± 95% CI for normally distributed data (**b**,**c**) in μmol/L. * *p* < 0.05; ** *p* < 0.01; *** *p* < 0.001. Abbreviations: SGA, small for gestational age; FGR, fetal growth restriction; IQR, interquartile range; CI, confidence interval.

**Figure 2 cells-11-03110-f002:**
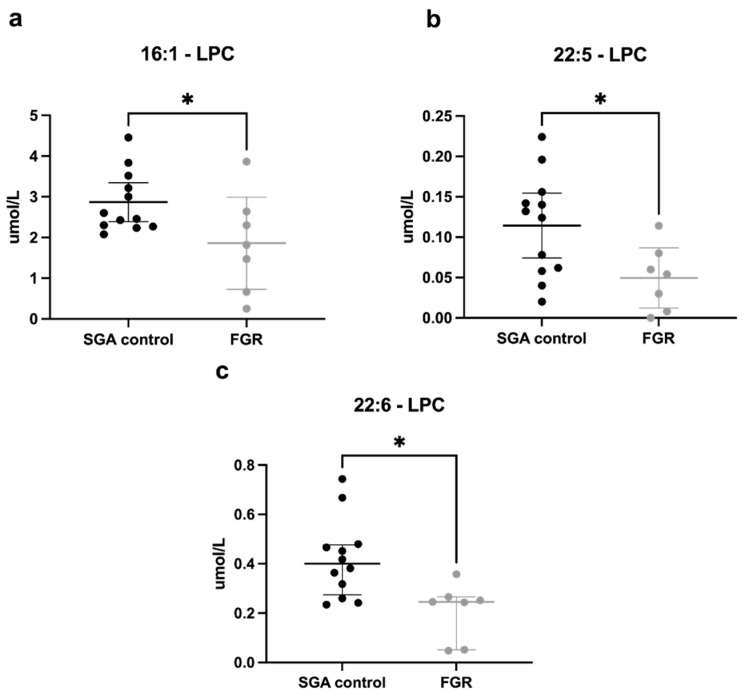
Lysophosphatidylcholine (LPC) concentrations (μmol/L) in umbilical vein plasma. SGA control (n = 12) and FGR (n = 7) umbilical vein plasma concentrations of LPC molecules. The fatty acid measured is that remaining attached to the glycerol backbone of a phosphatidylcholine (PC) molecule, following hydrolysis and removal of the second fatty acid. Significantly lower concentrations of LPC containing (**a**) palmitoleic acid (16:1), (**b**) docosapentaenoic acid (22:5), and (**c**) docosahexaenoic acid (22:6) were demonstrated in FGR plasma. Concentrations are expressed as mean ± 95% CI for normally distributed data (**a**,**b**) or median ± IQR for non-normally distributed data (**c**) in μmol/L. * *p* < 0.05. Abbreviations: SGA, small for gestational age; FGR, fetal growth restriction; CI, confidence interval; IQR, interquartile range.

**Figure 3 cells-11-03110-f003:**
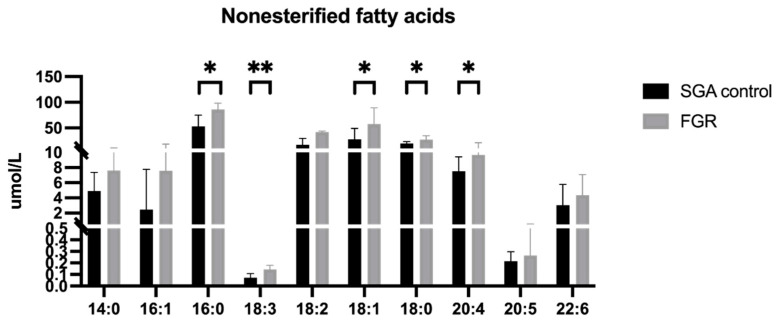
Non-esterified fatty acid concentrations (μmol/L) in umbilical vein plasma. SGA control (n = 12) and FGR (n = 7) umbilical vein plasma concentrations of individual non-esterified fatty acids, expressed as median ± IQR in μmol/L. * *p* < 0.05; ** *p* < 0.01. Abbreviations: SGA, small for gestational age; FGR, fetal growth restriction; IQR, interquartile range.

**Figure 4 cells-11-03110-f004:**
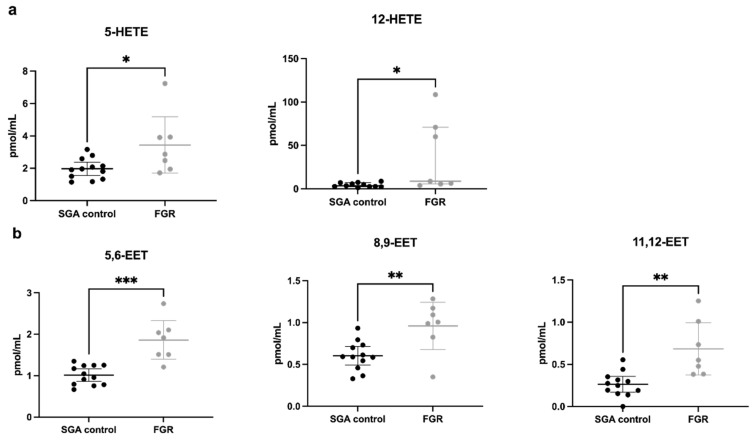
Eicosanoid concentrations (pmol/mL) in umbilical vein plasma. SGA control (n = 12) and FGR (n = 7) umbilical vein plasma concentrations of eicosanoids, arachidonic acid-derived lipid molecules. Concentrations of two families of eicosanoid molecules, (**a**) HETEs and (**b**) EETs were significantly elevated in FGR cord plasma. Concentrations are expressed as mean ± 95% CI (with the exception of 12-HETE, expressed as median ± IQR) in pmol/mL. * *p* < 0.05; ** *p* < 0.01; *** *p* < 0.001. Abbreviations: SGA, small for gestational age; FGR, fetal growth restriction; HETE, hydroxyeicosatetraenoic acid; EET, epoxyeicosatrienoic acid; CI, confidence interval; IQR, interquartile range.

**Figure 5 cells-11-03110-f005:**
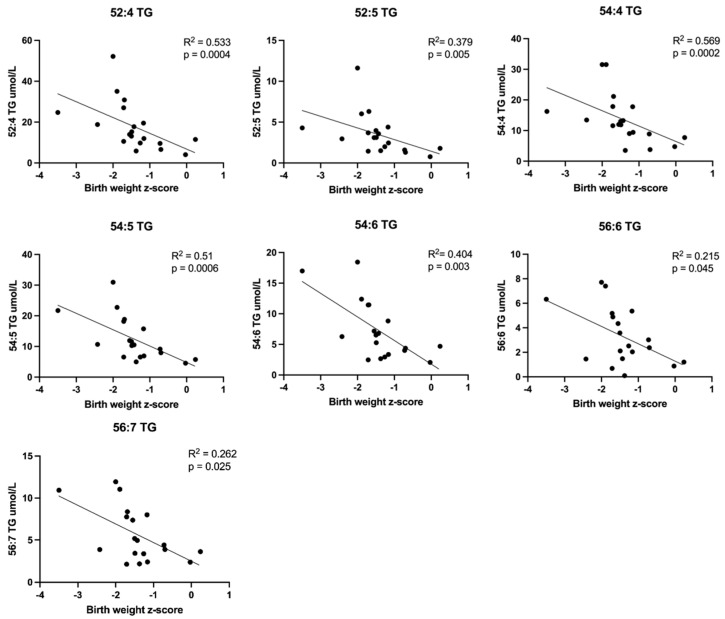
Correlations between umbilical vein plasma lipid concentrations and *birth weight z–score*: triglycerides. Scatterplots demonstrate the correlation relationships between birth weight z-score and plasma triglyceride concentrations containing higher numbers of carbon atoms and double bonds most likely to contain long chain polyunsaturated fatty acids. Spearman’s rank correlation was computed to assess relationships.

**Figure 6 cells-11-03110-f006:**
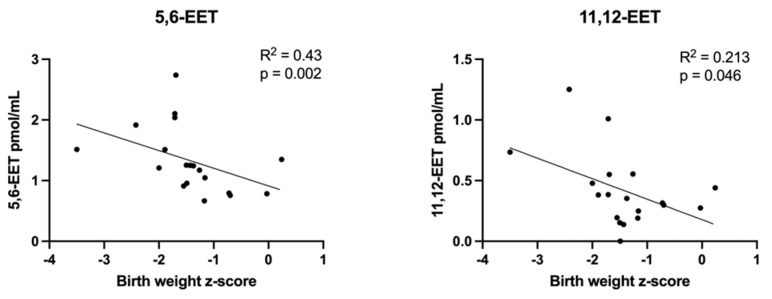
Correlations between umbilical vein plasma lipid concentrations and *birth weight z–score*: eicosanoids. Scatterplots demonstrate the correlation relationships between birth weight z-score and plasma eicosanoid epoxyeicosatrienoic (EET) acid molecules. Spearman’s rank correlation was computed to assess relationships.

**Figure 7 cells-11-03110-f007:**
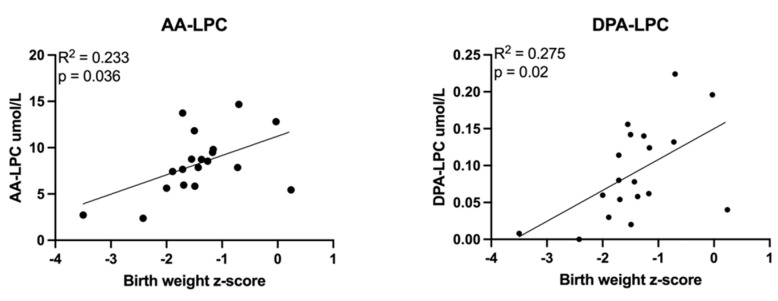
Correlations between umbilical vein plasma lipid concentrations and *birth weight z–scores*: lysophosphatidylcholines. Scatterplots demonstrate the correlation relationships between birth weight z-score and plasma lysophosphatidylcholine (LPC) molecules containing long chain polyunsaturated fatty acids. Abbreviations: AA, arachidonic acid; DPA, docosapentaenoic acid. Spearman’s rank correlation was computed to assess relationships.

**Figure 8 cells-11-03110-f008:**
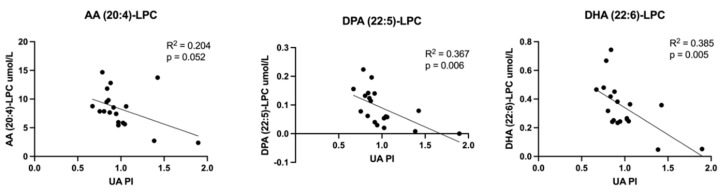
Correlations between umbilical vein plasma lipid concentrations and umbilical artery pulsatility index (UA PI): lysophosphatidylcholines. Scatterplots demonstrate correlation relationships between UA PI and umbilical vein plasma concentrations of lysophosphatidylcholine (LPC) molecules containing long chain polyunsaturated fatty acids. Abbreviations: AA, arachidonic acid; DPA, docosapentaenoic acid; DHA, docosahexaenoic acid. Pearson correlation coefficient was computed to assess relationships.

**Figure 9 cells-11-03110-f009:**
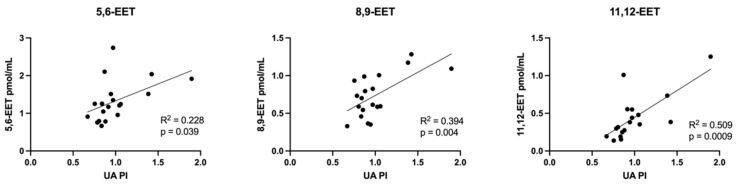
Correlations between umbilical vein plasma lipid concentrations and umbilical artery pulsatility index (UA PI): eicosanoids. Scatterplots demonstrate correlation relationships between UA PI and umbilical vein plasma concentrations of eicosanoid epoxyeicosatrienoic acid (EET) molecules. Pearson correlation coefficient was computed to assess relationships.

**Figure 10 cells-11-03110-f010:**
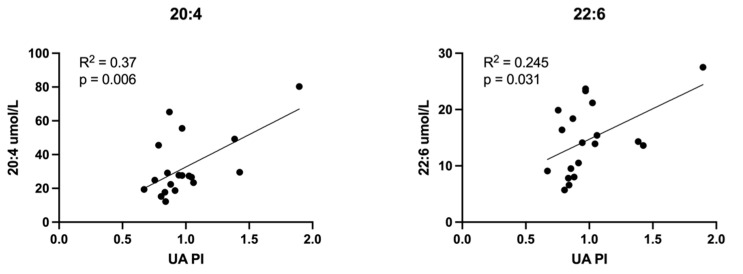
Correlations between umbilical vein plasma lipid concentrations and umbilical artery pulsatility index (UA PI): non-esterified fatty acids. Scatterplots demonstrate correlation relationships between UA PI and umbilical vein plasma concentrations of non-esterified long chain polyunsaturated fatty acids, arachidonic acid (20:4) and docosahexaenoic acid (22:6). Pearson correlation coefficient was computed to assess relationships.

**Figure 11 cells-11-03110-f011:**
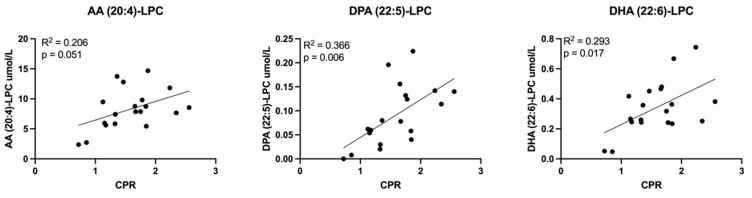
Correlations between umbilical vein plasma lipid concentrations and cerebroplacental ratio (CPR). Scatterplots demonstrate direct correlation between CPR (ratio of MCA PI divided by UA PI) and umbilical vein plasma concentrations of long chain polyunsaturated fatty acid (LCPUFA)-lysophosphatidylcholine (LPC) molecules. Abbreviations: AA, arachidonic acid; DPA, docosapentaenoic acid; DHA, docosahexaenoic acid; MCA PI, middle cerebral artery pulsatility index; UA PI, umbilical artery pulsatility index. Pearson correlation coefficient was computed to assess relationships.

**Figure 12 cells-11-03110-f012:**
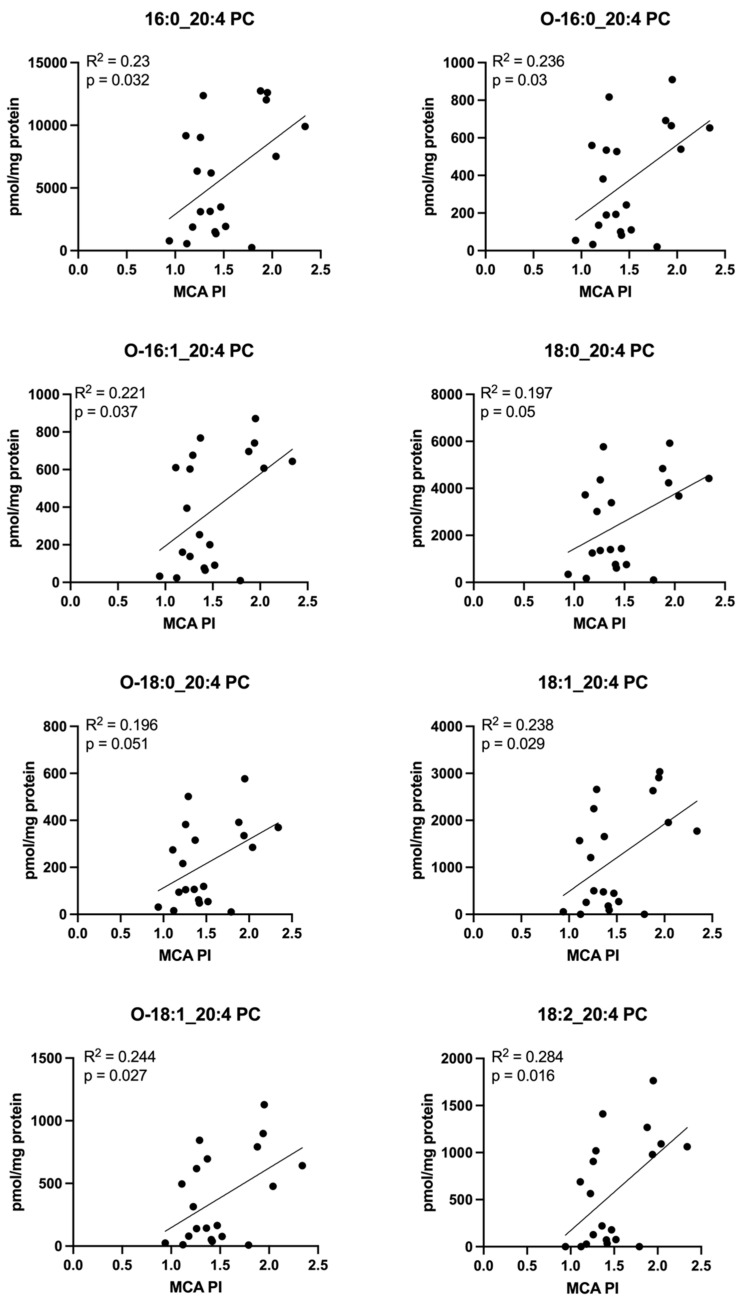
Correlations between MCA PI and concentrations of phosphatidylcholine molecules containing arachidonic acid (20:4) in the placenta. Scatterplots demonstrate direct correlation between middle cerebral artery pulsatility index (MCA PI) and placental concentrations of multiple phosphatidylcholine (PC) molecules containing arachidonic acid (20:4). “O-” denotes an ether linkage. Pearson correlation coefficient was computed to assess relationships.

**Figure 13 cells-11-03110-f013:**
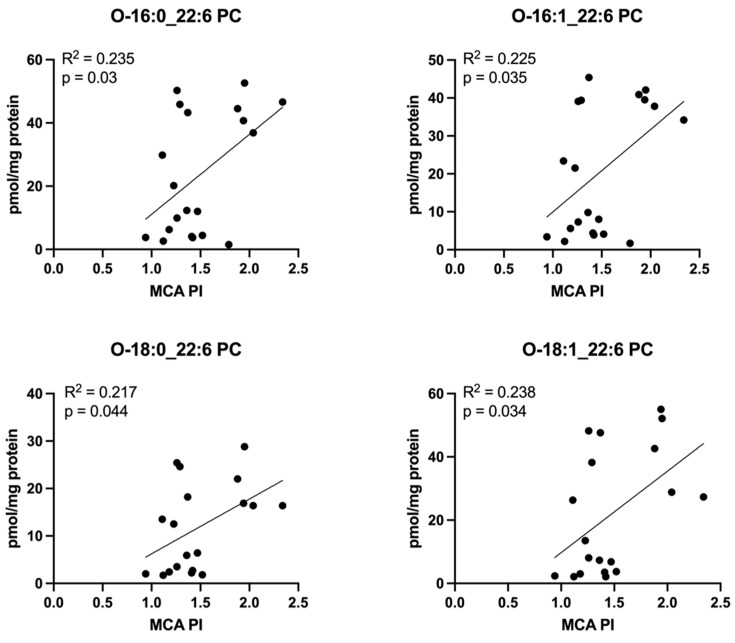
Correlations between MCA PI and concentrations of phosphatidylcholine molecules containing docosahexaenoic acid (DHA, 22:6) in the placenta. Scatterplots demonstrate direct correlation between middle cerebral artery pulsatility index (MCA PI) and placental concentrations of multiple phosphatidylcholine (PC) molecules containing DHA (22:6). “O-” denotes and ether linkage. Pearson correlation coefficient was computed to assess relationships.

**Table 1 cells-11-03110-t001:** Clinical characteristics and ultrasound measurements of SGA Controls and FGR Cases.

Characteristic	SGA Controls (n = 12)		FGR Cases (n = 8)		*P* Value
	Mean/Median	95% CI/IQR	Mean/Median	95% CI/IQR	
**Maternal, n (%)**					
Age (years)	29.25	[25.3,33.2]	27.13	[23.1,31.1]	0.426
Race					0.495
White	10 (83%)		8 (100%)		
Not reported	2 (17%)				
Chronic hypertension					>0.999
Yes	1 (8%)		1 (12%)		
No	11 (92%)		7 (88%)		
PIH					
No	12 (100%)		8 (100%)		
Preeclampsia					>0.999
Yes	1 (8%)		0		
No	11 (92%)		8 (100%)		
Mode of delivery					0.255
Vaginal	11 (92%)		5 (62%		
C-section	1 (8%)		3 (38%)		
**Prenatal, n (%)**					
GA at recruitment					>0.999
<32 weeks	5 (42%)		4 (50%)		
	7 (58%)		4 (50%)		
**Ultrasound measurements ***					
GA at ultrasound (weeks)	36.57	[36.1,37.1]	34.11	[31.1,37.1]	0.032
EFW (grams)	2403	[2263,2543]	1713	[1121,2305]	0.005
EFW percentile (%)	10.5	[16.5]	1.5	[3.5]	0.008
AC (cm)	31.18	[2.03]	26.84	[7.2]	0.069
AC percentile (%)	24.83	[10.2,39.5]	2.5	[0.7,4.3]	0.015
UA PI	0.87	[0.8,0.9]	1.16	[0.8,1.5]	0.019
UA PI percentile (%)	59.5	[47.5,71.5]	78.67	[60.8,96.6]	0.049
MCA PI	1.53	[1.3,1.8]	1.45	[1.2,1.7]	0.641
MCA PI percentile (%)	16.15	[34.1]	11.23	[25.6]	0.384
CPR	1.76	[1.5,2]	1.35	[0.9,1.8]	0.060
CPR percentile (%)	34.46	[42.6]	4.61	[37.1]	0.083
**Birth**					
Infant sex, n (%)					>0.999
Male	4 (33%)		2 (25%)		
Female	8 (67%)		6 (75%)		
GA at delivery (weeks)	38	[37.4,38.7]	36.52	[34.8,38.2]	0.038
Placenta weight (grams)	535.30	[472.9,597.7]	374.40	[242.4,506.4]	0.01
Birth weight (grams)	2590	[467]	2120	[600]	0.0002
Birth weight percentile (%)	11	[16.8]	3.5	[2.8]	<0.0001
Birth length (cm)	46.97	[46.4,47.6]	42.54	[38,47.1]	0.012
Birth length percentile (%)	21	[20]	7.5	[12.3]	0.02
Birth HC (cm)	32.82	[31.9,33.8]	31.64	[29.9,33.4]	0.143
Birth HC percentile (%)	15.50	[50.3]	12	[40]	0.521
HC:BW ratio	12.73	[2.5]	14.88	[1.5]	0.0001
**Z** **–scores**					
Birth weight	−1.215	[0.8]	−1.80	[0.6]	<0.0001
Birth length	−0.79	[−1.2,−0.4]	−1.80	[−2.9,−0.8]	0.027
Birth HC	−0.55	[−1.1,−0.02]	−0.97	[−1.7,−0.2]	0.3

* measurements from final ultrasound prior to delivery; Demographics and clinical characteristics were analyzed for normal distribution. Normally distributed continuous variables were compared using unpaired t-test and are expressed as mean [95% confidence interval]. Non-normally distributed continuous variables were compared using Mann-Whitney test and are expressed as median [interquartile range]. Categorical variables were compared used Fisher’s exact test. Abbreviations: SGA, small for gestational age; FGR, fetal growth restriction; CI, confidence interval; IQR, interquartile range; PIH, pregnancy induced hypertension; GA, gestational age; EFW, estimated fetal weight; AC, abdominal circumference; UA, umbilical artery; PI, pulsatility index; MCA, middle cerebral artery; CPR, cerebroplacental ratio; HC, head circumference; BW, birth weight.

## Data Availability

Any data that support these findings and are not included within the main article and its Appendix A are available from the corresponding author upon reasonable request.

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
