# Peer review of "Altered Cord Blood Lipid Concentrations Correlate with Birth Weight and Doppler Velocimetry of Fetal Vessels in Human Fetal Growth Restriction Pregnancies"

_cells, 2022, doi:10.3390/cells11193110_

Round 1

Reviewer 1 Report

GENERAL REMARKS

This study compares LCPUFA’s in umbilical vein and placenta with birth weight and fetal Doppler measurements. My major objective is the selection of the two groups that are compared – see arguments below. The lack of a proper control group hampers the comparisons, as is written by the authors in the discussion. I would suggest to restrict the presentations to the associations of LCPUFA’s and birth weight, or possibly birth weight z-score or birthweight MoM, and fetal Dopplers. Adjustment to GA could be considered. Then you evade the problem that reliable differentiation between SGA and FGR is impossible. You might also restrict the number of molecules that you show and move the remaining ones to the addendum.

INTRODUCTION

Line 54: “secondary to increased placental vascular resistance” – I always thought that the most frequent primary cause of placental insufficiency is reduced maternal vascular supply due to deficient adaptation of the spiral arteries, and that increased placental resistance is secondary to that. This is what you describe after this sentence.

MATERIAL – METHODS

Line126: You recruited women between 2015-2019, but your study population includes only 20 women. I suppose that these were not all women with EFW <p10 in your clinic during these 4 years, so what further selection did you use?

Line 163: Your classification seems only differentiates more severe from less severe case of FGR. You have no proof that p5-p10 is SGA and <p5 is FGR. I wouldn’t consider >p5 to p10 as “normal controls”

Line 225: for t-test you have to be sure that the data are normally distributed, which often is not so in obstetric data (e.g. low Apgar scores are far less frequent then normal scores and therefore distribution is skewed). Presenting data as mean and standard error of the mean instead of standard deviation gives less information on the measurement variation.

RESULTS

Because the group classification based on percentiles is not precise, you might delete the figures showing fatty acids compared between groups and only the figures that show the association with BW and Doppler. In this way the large number of figures is reduced, which might make it easier to follow the text. Instead of BW you could use a multiple of the median BW ratio = BW / median normal BW for GA, or Z-scores. This presents better information on the severity of FGR. Furthermore, you may have to adjust for GA, as fatty acid concentration in umbilical vein is dependent on GA.

DISCUSSION

The first sentence is more apt for the introduction. The 2nd sentence is not so interesting that you should start the discussion with this. Usually it is best to start the discussion with your main findings and how to interpret these.

Line 421: I do not understand your hypothesis. Decrease of some lipid molecules and increase of others in FGR was described earlier, with similar results to your study, although earlier studies did not include Doppler.

Line 430: “Additionally, a fetus with higher UA PI demonstrates high levels of EET molecules and nonesterified LCPUFA (DHA, arachidonic acid) and low levels of LCPUFA-LPCs (arachidonic acid, DPA, DHA).” – I do not understand this sentence, you mention the same molecule twice with different characteristics.

Line 433: “Differentiating pathological growth in FGR fetuses from constitutionally small fetuses is challenging”. I agree with this statement. However, your data are unconvincing. I would expect normal values of PUFA in genetically small babies (or SGA) who had continuous growth at a lower velocity. If you had data on normal pregnancies you could have substantiated this. FGR presents as a continuum from severe FGR to nearly normal growth. Your data show that more severe FGR differs from less severe FGR. Doppler helps to differentiate severe FGR from less severe – in late FGR Doppler is often within the normal range.

Line 450: “Seven of the eight subjects in this FGR group also met the Delphi consensus group criteria for FGR diagnosis” Which Doppler chart did you use for this calculation. Although the Consensus Definition seems effective for differentiating FGR from SGA or lesser case of FGR, the choice of reference chart for fetal weight and Doppler measurements has far more impact on selection than the choice between SMFM definition (<p10) or Consensus Definition.

Line 469: “The storage of fatty acids in triglycerides and lipid droplets under hypoxic conditions is suspected to be a protective response to prevent intracellular lipotoxicity, buffering against the known lipotoxic properties of nonesterified fatty acids” – any reference for this statement?

Reviewer 2 Report

Chassen et al. presented the study to explore the correlation between fetoplacental lipidomic profiles and measures of fetal growth and wellbeing. They identified that, under the condition of fetal growth restriction in utero, placental metabolism and fetal lipoprotective strategies will be altered. The method is straightforward and the figures are easy to follow. This paper addressed a significant clinical problem, and the results will be insightful for the treatment of the target population. My major concern in this study is that the data interpretation is weak and the conclusion could not be fully supported based on the current context.

Following are my major and minor comment points.

Major points:

1.     Please correct your supplementary appendices numbering. Sup A-J does not exist and I cannot match your in-text Sup tables with your appendices.

2.     In line 307-309, the authors claimed the inverse correlation between triglycerides and BW. However, in Fig. 6, only “52:4TG” showed statistical significance. Please explain.

3.     For correlation analysis in this study, r-square is suggested to be included to examine the variance in the chart.

4.     In line 307-310, LCPUFA in the triglyceride species group and LPCs group showed opposite correlation. Please explain.

5.     In line 342, the correlation between LCPUFA and LPCs was analyzed while other groups were excluded(such as triglyceride species). Please explain.

6.     In line 413, “To our knowledge, this is the first study to explore fetal and placental lipid 413 concentrations and their relationship with ultrasound Doppler assessments in FGR”. There are previous studies explored this relationship. Please see the doi below:

https://doi.org/10.1038/s41598-018-31832-5

doi:10.1186/s12884-020-2753-1

7.     Considering the small sample size and confounding factors, the limitations of this study should be better addressed and a more objective conclusion is suggested.

Minor Points:

8.     In the methods section, the manufacturers or brands of the instruments, including Doppler, should be listed.

9.     In the tissue collection, a standardized protocol was mentioned. Where is this protocol from? References will be suggested.

10.  In line 461, please cite your reference for the increased triglyceride concentrations suggested hypoxia.

11.  Statistical Analysis: Please state whether data have been analyzed for normality and equal variance as a justification for using parametric analysis.

12.  Please state the type of statistical test applied to each dataset. If different statistical tests are used within the manuscript, the specific test used should be stated in the Methods, Results and Figure legend.

Reviewer 3 Report

This study is simple because by using HPLC analysis and Doppler you get a lot of data that you can show in many different way. Although the study is simple, it is generally well written and new (which are the strengths of this paper). If I would be picky it would be useful reducing the number of figures to make the manuscript easy to read. Moreover, authors can highlight the small sample size which is a weakness of this study but I guess that it is not possible to increase the cohort so it would be useful stating this weakness. Overall is a good paper.

Round 2

Reviewer 2 Report

The authors have addressed most of my comments. 

For the current version, I would suggest the author include the unpublished protocol for sample collection and processing.  This is important for the reproducibility of the research and transparency of the project. 

Author Response

Reviewer #2 comment: For the current version, I would suggest the author include the unpublished protocol for sample collection and processing.  This is important for the reproducibility of the research and transparency of the project. 

Response:

We thank you once again for your great attention to detail. We have revised the manuscript to elaborate a bit on the details of the placenta collection guideline that our lab followed for this study. As detailed beginning on line 155 in the revised version of our manuscript, research groups conducting studies on placentas from FGR pregnancies during our study period were allowed to take 10% of the total placental weight. This guideline was put into place at the discretion of the Director of OBGYN Pathology at the time, in order to ensure that enough placental tissue remained for clinical pathologic evaluation and clinical management of mom and FGR infant. Unfortunately a formal document outlining this placenta collection procedure is not available/does not exist, but we clarified the reasoning in the Methods section as referenced above. The details of the processing/homogenization of the placenta have been provided and expanded on within the same Methods paragraph.